# Greater Muscular Strength Is Associated with a Lower Risk of Pulmonary Dysfunction in Individuals with Non-Alcoholic Fatty Liver Disease

**DOI:** 10.3390/jcm11144151

**Published:** 2022-07-17

**Authors:** Jinkyung Cho, Bruce D. Johnson, Kymberly D. Watt, Chul-Ho Kim

**Affiliations:** 1Department of Cardiovascular Disease, Mayo Clinic, Rochester, MN 55905, USA; chojk@kspo.or.kr (J.C.); johnson.bruce@mayo.edu (B.D.J.); 2Department of Sports Science, Korea Institute of Sport Science, Seoul 01794, Korea; 3Department of Gastroenterology and Hepatology, Mayo Clinic, Rochester, MN 55905, USA; watt.kymberly@mayo.edu

**Keywords:** non-alcoholic fatty liver disease, pulmonary function, handgrip strength

## Abstract

This study investigated the combined effect of handgrip strength (HGS) and non-alcoholic fatty liver disease (NAFLD) on pulmonary function using the Korea National Health and Nutrition Examination Survey (KNHANES) from 2016 to 2018. For the present study, forced vital capacity (FVC), forced expiratory volume in 1 second (FEV_1_), the FEV_1_/FVC ratio, handgrip strength (HGS) and the hepatic steatosis index (HSI) to estimate NAFLD were obtained from nationwide cross-sectional surveys. For HGS, subjects were divided into higher HGS (upper 50%) and lower HGS (lower 50%). For NAFLD, subjects were divided into the NAFLD cohort (HSI > 36) and the normal cohort (HSI ≤ 36). Of 1651 subjects (men, *n* = 601), 25.5% of subjects (*n* = 421) met the HSI > 36. Based on the normal cohort with high HGS, the normal cohort with low HGS showed an increased risk of reduced FVC (OR = 3.062, 95% CI = 2.46–4.83, *p* < 0.001) and the NAFLD cohort with low HGS showed a further increased risk of reduced FVC (OR = 4.489, 95% CI = 3.43–7.09, *p* < 0.001). However, the risk of reduced FVC was not significantly increased in NAFLD with high HGS (OR = 1.297, 95% CI = 0.67–2.50, *p* = 0.436). After adjusted for covariates such as age, sex, smoking, FBG, HDL-C, TG, SBP, DBP, CRP and alcohol consumption, the results remained similar. More importantly, these results were consistent in the obesity-stratified analysis. The current findings of the study suggest that higher muscle strength is associated with a lower risk of reduced pulmonary function in individuals with NAFLD.

## 1. Introduction

Non-alcoholic fatty liver disease (NAFLD) is a broad term for a wide range of liver conditions linked to metabolic syndrome in which excessive fat is accumulated with little or no alcohol consumption. NAFLD is the most common type of liver disease globally and the prevalence has been continuously increasing [1,2]. Approximately 25% of adults worldwide have NAFLD and many of the cases develop into more severe cases of NAFLD, including non-alcoholic steatohepatitis (NASH) or cirrhosis [2,3]. Furthermore, NAFLD is linked to an increase in all-cause mortality relating to hepatic and non-hepatic complications [4]. In addition, NALFD has appeared to be related to several types of extra hepatic pulmonary complications such as chronic obstructive pulmonary disease (COPD), asthma, obstructive sleep apnea (OSA), pneumonia, pulmonary arterial hypertension (PAH), pulmonary fibrosis and pulmonary cancer [5,6]; moreover, COPD, pulmonary fibrosis and PAH are the main causes of death in NAFLD [5]. As such, pulmonary disease appears as one of the major “cause-specific death” and “contributory causes of death” in NAFLD [7]. NAFLD is also closely associated with pulmonary dysfunction in which pressure, blood flow, compliance and volume are involved, and patients with NAFLD often demonstrate a reduction in pulmonary function (e.g., lung mechanics and gas exchange). Alteration in pulmonary pathophysiology is an established clinical complication associated with NAFLD. NAFLD was independently associated with impaired pulmonary physiology with a more severe degree of hepatic disease, associating it with the greatest risk for poor pulmonary function [8,9]. Severe pulmonary abnormalities that are well established with cirrhosis and portal hypertension include portopulmonary hypertension and hepatopulmonary syndrome, which occur by a different physiologic route and are not specific to NAFLD [6,10].

Handgrip strength (HGS) is widely assessed in public health since it is independently associated with clinical disorders [11,12,13]. In addition, it is a common indicator of general muscular strength [14] and has been indirectly linked with respiratory muscle strength and function as well as general vigor, and may represent a general form of frailty [15,16]. Given this association of HGS with respiratory muscle function, it follows that it is also positively related to measures of pulmonary function [17]. Therefore, the purpose of the present study was to investigate the combined effect of NAFLD and HGS on pulmonary function and their association in a large cohort of individuals from the Korea National Health and Nutrition Examination Survey (KNHANES) from 2016 to 2018.

## 2. Materials and Methods

### 2.1. Study Participants

The KNHANES is a nationwide, population-based, cross-sectional health and nutrition survey conducted by the Korea Centers for Disease Control and Prevention (KCDC) to monitor health and nutritional status in Korea [18]. Figure 1 shows the flowchart of participant selection. In this study, 16,489 potential subjects were initially considered. Since spirometry was examined only in participants aged 40 years and older, 2230 subjects were selected for the analysis. In addition, subjects who had following criteria were excluded from the present study (Figure 1): (1) absence of aspartate aminotransferase (AST) and alanine aminotransferase (ALT) data (*n* = 68), (2) absence of HGS data (*n* = 130), (3) liver diseases including hepatitis and cancer (*n* = 37), (4) alcohol consumption per week > 140 g for men (*n* = 294) and >70 g for women (*n* = 50). Consequently, 579 were excluded and a total of 1651 subjects (*n* = 601 in men) were included in the analysis.

### 2.2. Covariates

Obesity was defined according to the Korean Society for the Study of Obesity and a cut-off point of body mass index (BMI) ≥ 25 kg/m^2^ was applied [19]. BMI was determined by weight (kg)/height^2^ (m^2^). A self-reported questionnaire was used to assess age, sex and health-related factors (smoking, alcohol consumption, hypertension and diabetes). Smoking was classified “yes” if the individuals had smoked 5 packs of cigarettes or more in their lifetimes [20]. Fasting blood glucose (FBG), total cholesterol (TC), high-density lipoprotein cholesterol (HDL-C), triglyceride (TG), AST, ALT and C-reactive protein (CRP) levels were measured using the Hitachi Automatic Analyzer 7600-210 (Hitachi, Tokyo, Japan). Alcohol consumption was classified “yes” if the individuals had ≥2 alcoholic drinks per week [21]. Hypertension was defined as blood pressure ≥ 130/85 mmHg or by the usage of antihypertensive medications [22]. Diabetes was defined as FBG ≥ 126 mg/dL or by the diagnosis of a physician [23].

### 2.3. Pulmonary Function Measurements

The pulmonary function test was performed using spirometry (2130 Spirometer, Sensor Medics, Yorba Linda, CA, USA) by well-trained technicians following the guidelines from the American Thoracic Society [24]. Forced vital capacity (FVC) and forced expiratory volume in 1 second (FEV_1_) were measured at least 3 times and the highest values were used. Predicted FVC and FEV_1_ were expressed as FVC (%pred.) and FEV_1_ (%pred.), respectively. The values were determined using the Korean reference equations based on representative samples of the Korean population [25]. The equations are as follows:

FVC (%pred.)
Male: −4.8434 − 0.00008633*age (years)^2^ + 0.05292*height (cm) + 0.01095*body weight (kg)
Female: −3.0006 − 0.0001273*age (years)^2^ +0.03951*height (cm) + 0.006892*body weight (kg)FEV_1_ (%pred.)
Male: −3.4132 − 0.0002484*age (years)^2^ + 0.04578*height (cm)
Female: −2.4114 − 0.0001920*age (years)^2^ + 0.03558*height (cm)

For analysis, FVC (%pred.) < 80% and/or FEV_1_ (%pred.) < 80% was categorized as poor pulmonary function.

### 2.4. Definition of NAFLD

To define NAFLD subjects, we used the hepatic steatosis index (HSI), a non-invasive algorithm based on laboratory and anthropometric variables to estimate the risk of NAFLD. This algorithm was validated [26,27] and widely applied in the current literature [28,29]. The HSI was calculated as the following formula: 8 (alanine transaminase [ALT]/aspartate transaminase [AST]) ratio + BMI (+2 if diabetes mellitus [DM] status + 2 if female)

Values > 36 were considered to present NAFLD.

### 2.5. Relative Handgrip Strength

The handgrip strength of the dominant hand (HGS) was measured three times in a standing position using a digital hand dynamometer (digital grip strength dynamometer, T.K.K. 5401, Takei Scientific Instruments Co., Ltd., Tokyo, Japan). The maximum value of the measured values was chosen as absolute HGS. The relative HGS was computed as absolute HGS divided by BMI. The relative HGS was classified as either high HGS (upper 50%) or low HGS (lower 50%), according to sex.

### 2.6. Statistical Analysis

Statistical analysis was conducted using SPSS software (Version 24.0 for Windows, SPSS, Inc., Chicago, IL, USA) and *p*-values < 0.05 were considered statistically significant. All the variables were checked for normality, and if necessary, subjected to log10 transformation before analysis. Data are presented as mean ± standard deviation (SD) for continuous variables and as proportions (%) for categorized variables. The independent t-test, one-way ANOVA and chi-square test were used to compare the mean differences according to NAFLD and the HGS subgroups. Univariate logistic analysis was conducted to determine the risk (odds ratio (OR)) of reduced pulmonary function for the NAFLD cohort and HGS. Binary logistic analysis was conducted to determine the risk (OR) of reduced pulmonary function for the combination of NAFLD and HGS after adjusting for age, sex, FBG, HDL-C, TG, SBP, DBP, CRP, smoking and alcohol consumption. Individuals who were in the normal cohort and had a high HGS were used as the reference (OR = 1).

## 3. Results

Of the 1651 subjects that included 601 men and 1501 women in this study, 421 (25.5%) met the index for NAFLD using cut-off values of HSI > 36. The NAFLD cohort had more adverse blood markers and health-related factors than the normal cohort. Values of BMI, AST, ALT, FBG, TG, SBP, DBP and CRP, and frequencies of hypertension and diabetes were significantly higher in the NAFLD cohort (Table 1). Regarding the spirometric values, the FVC (%pred.) value was significantly lower in the NAFLD cohort (*p* < 0.001), whereas FEV_1_/FVC was significantly higher in the NAFLD cohort (*p* < 0.001) than in the normal cohort. There was no difference in FEV_1_ (%pred.) between the NAFLD cohort and the normal cohort (*p* > 0.05).

Subjects were classified according to relative HGS (Table 2). Age, BMI, AST, ALT, FBG, TG, SBP and CRP, and frequencies of hypertension and diabetes were all significantly higher in the low HGS group. The HSI was significantly higher in the low HGS group (*p* < 0.001). With respect to pulmonary function parameters, FEV_1_ (%pred.), FVC (%pred.) and FEV_1_/FVC were significantly lower in the low HGS group compared to the high HGS group (*p* = 0.031, *p* < 0.001 and *p* = 0.001).

A univariate logistic regression analysis was performed to clarify the effect of NAFLD or relative HGS on pulmonary function. When the risk of FVC (%pred.) < 80%, the risk was significantly increased in the NAFLD cohort than the normal reference (OR = 1.938, 95% CI = 1.48–2.54, *p* < 0.001, Table 3) and it was increased in the low HGS group compared to the high HGS group (OR = 1.951; 95% CI = 1.69–2.26, *p* < 0.001, Table 3). When the risk of FEV_1_ (%pred.) < 80%, the risk was not significantly changed in the NAFLD cohort compared to the normal reference (OR = 0.952, 95% CI = 0.71–1.28, *p* = 0.747, Table 3) whereas it was increased in the low HGS group compared to the high HGS group (OR = 1.507, 95% CI = 1.16–1.96, *p* = 0.002, Table 3).

Table 4 presents results of the combined effect of NAFLD and HGS on the risk of reduced pulmonary function. Compared to the normal cohort with high HGS (reference), the normal cohort with low HGS had an increased risk of FVC (%pred.) < 80% (OR = 3.062, 95% CI = 2.46–4.83, *p* < 0.001) and the NAFLD cohort with low HGS had a further increased risk of FVC (%pred.) < 80% (OR = 4.489, 95% CI = 3.43–7.09, *p* < 0.001). After adjusting for covariates such as age, sex, smoking, FBG, HDL-C, TG, SBP, DBP, CRP and alcohol consumption, the risk of FVC (%pred.) < 80% was still increased in the normal cohort with low HGS (OR = 2.036, 95% CI = 1.40–2.97, *p* < 0.001) and the NAFLD cohort with low HGS (OR = 3.390, 95% CI = 2.23–5.15, *p* < 0.001).

The subsequent analysis was conducted to observe the combined effect of NAFLD and HGS on the risk of reduced pulmonary function according to the presence or absence of obesity. The obesity-stratified analysis showed that in the non-obese population, the normal cohort with low HGS had an increased risk of FVC (%pred.) < 80% (OR = 2.480, 95% CI = 1.51–4.07, *p* < 0.001) and the NAFLD cohort with low HGS had a further increased risk of FVC (%pred.) < 80% (OR = 3.856, 95% CI = 1.49–9.99, *p =* 0.005) compared to the normal cohort plus high HGS (reference) after adjusting for all covariates. Similarly, in the obese population, compared to the normal cohort with high HGS (reference), the NAFLD with low HGS had an increased risk of FVC (%pred.) < 80% (OR = 2.192, 95% CI = 1.15–4.16, *p* = 0.017) after adjusting for all covariates.

## 4. Discussion

In the present study, the NAFLD cohort had lower HGS and pulmonary function than the normal cohort. Individuals with low HGS showed more severe NAFLD-related risk factors and impaired pulmonary function than those with high HGS. In addition, the NAFLD cohort demonstrated a restrictive patterned pulmonary dysfunction, as indicated by reduced FVC, normal FEV_1_ and a higher FEV_1_/FVC compared to the normal cohort. For the combined effect of NAFLD and HGS on pulmonary function, incremental increases in the risk of reduced pulmonary function were observed in the normal cohort with low HGS (~3 folds) and NAFLD cohort with low HGS (~4.5 folds). However, the risks of reduced pulmonary function could be decreased in NAFLD if HGS was high. More importantly, this result was consistent with and without obesity. This phenomenon was not altered after adjusting for other confounding factors.

The positive relationship between NAFLD and pulmonary dysfunction has been well established by different types of observations, including a cross-sectional study using the data from the third National Health and Nutrition Examination Survey (NHANES III) involving 9976 participating adults [9], meta-analyzing data from five cross-sectional studies and one longitudinal study involving 133,707 individuals [30], and a prospective study using liver biopsy [31]. Likewise, the present study observed reduced pulmonary function in the NAFLD cohort. Pulmonary dysfunction is a secondary health complication in NAFLD. The underlying mechanism relating to pulmonary dysfunction in NAFLD is still not clearly understood. In the present study and another previous study [32], a smoking history that could directly affect pulmonary function was not the major factor for the reduced pulmonary function in NAFLD. As such, it has been hypothesized that the reduced pulmonary function in NAFLD results from an increase in inflammatory and/or circulating mediators eliciting pathophysiological and histological changes in the parenchymal and the porto-pulmonary system [33,34,35,36]. However, despite this pathophysiological link between the liver and the lungs, the pulmonary function seems to be relatively maintained if muscular strength is greater. In the present study, a high HGS appeared to reduce the risks of reduced pulmonary function in NAFLD and, moreover, HGS seemed to have a greater impact on pulmonary function than NAFLD. This indicates that respiratory muscle strength likely plays an important role in pulmonary function in this circumstance. Although there are some limitations, this hypothesis is supported by previous studies reporting that fat free mass (or lean body mass) was a significant predictor of pulmonary function in the elderly [37], and peripheral muscle strength, respiratory muscle strength and respiratory muscle function develop simultaneously to each other and are associated with pulmonary function in healthy individuals [15]. Furthermore, the loss of skeletal muscle mass was independently correlated with a reduction in pulmonary function [38]. Accordingly, these results emphasize the importance of muscular strength training to maintain pulmonary function in patients with NAFLD and/or to attenuate an NAFLD-induced decline in pulmonary function. In public health, high-intensity strength training programs appear to be well-tolerated and improve pulmonary function in other pulmonary disease groups, including chronic obstructive pulmonary disease (COPD) [39,40]. In addition to muscular strength, it can be expected to improve risk factors of metabolic syndrome such as insulin resistance and hyperlipidemia [41,42].

Notwithstanding, mechanisms resulting in a loss of lung volume in the NAFLD cohort patients also contribute to pulmonary dysfunction. Lung volume, in this case FVC, is a balance of respiratory muscle strength, compliance of lung tissue and recoil properties of the chest wall. Thus, improving muscle strength should potentially impact this measure; however, smaller, more mechanistic studies suggest an impact of mediators in NAFLD, e.g., inflammatory pathways that may act directly on lung parenchyma or airway smooth muscle. However, the present study suggests that at least in this mild NAFLD population, muscle strength is important.

A limitation of the study includes whether the severity of pulmonary dysfunction can be measured accurately with spirometry. Indeed, a recent cross-sectional and longitudinal study revealed that lower FVC and FEV_1_ were related to NAFLD; however, the rates of decline were not related to NAFLD [43]. Other studies have found that pulmonary function was inversely associated with NAFLD severity [8,9]. Although more studies are needed, FVC and FEV_1_ may provide only limited information on pulmonary function in NAFLD, and a more advanced series of assessments may be required to understand pulmonary function and pathogenesis.

Importantly, the NAFLD cohort showed lower FVC relative to the normal cohort; however, the decline was relatively mild in the present study. In addition, FEV_1_ was not significantly different between the NAFLD cohort and the normal cohort. These findings are consistent with a recent study by Peng at al. that investigated the relationships among pulmonary function and patterns in NAFLD [9]. They found that FVC and FEV_1_ were significantly associated with the degree of NAFLD, but this result is limited to only moderate to severe cases of NAFLD. In addition, this study revealed that FVC was also significantly reduced without a significant change in FEV_1_ in mild cases of NAFLD. Therefore, it seems clear that FVC as a marker of general pulmonary function is reduced in patients with mild NAFLD. However, in the case of mild NAFLD, it is more likely that reduced FVC may be due to muscle weakness, perhaps related to de-conditioning from a lack of physical activity and a sedentary lifestyle rather than a pathophysiological process solely linked to NAFLD. As such, it is noted that the current finding is limited to the mild degree of NAFLD, and further studies are needed to explore the relationship between HGS and pulmonary function in more severe cases of NAFLD, as well as needing additional measures of pulmonary function beyond lung mechanics.

The present study has other limitations. First, it is difficult to verify the cause and effect regarding the relationship between NAFLD plus HGS and pulmonary function due to the cross-sectional nature of the study. Second, although liver biopsy is the gold standard to evaluate NAFLD, it was not available in this study. We used the HSI as a non-invasive diagnostic algorithm to define NAFLD. The HSI has been well-validated against liver biopsy using a large-scale, population-based study conducted in Korea [44]. Therefore, the NAFLD index using the HSI can be used to predict the risk of NAFLD in the Korean population. Third, although HGS was an independent variable for the risk of pulmonary function decline, body composition, such as fat free mass, was not considered. This study used the Korean database, which may or may not translate to other ethnic groups. This population was female dominant and the lower BMI in this population may reflect different effects on pulmonary dynamics than higher BMI populations. Finally, there was an age difference between low HGS and high HGS groups, and this might have influenced the results.

## 5. Conclusions

A population-based, cross-sectional study found that the NAFLD cohort demonstrated a lower FVC, a similar FEV_1_ and a higher FEV_1_/FVC relative to the normal cohort. The risk of reduced pulmonary function was associated with NAFLD; however, this reduction in pulmonary function appeared to decline more significantly relative to HGS. The results of the present study emphasize the importance of general muscle strength and, thus, physical training as a therapeutic strategy for patients with NAFLD. However, these findings may be applied at the present time to patients with milder NAFLD.

## Figures and Tables

**Figure 1 jcm-11-04151-f001:**
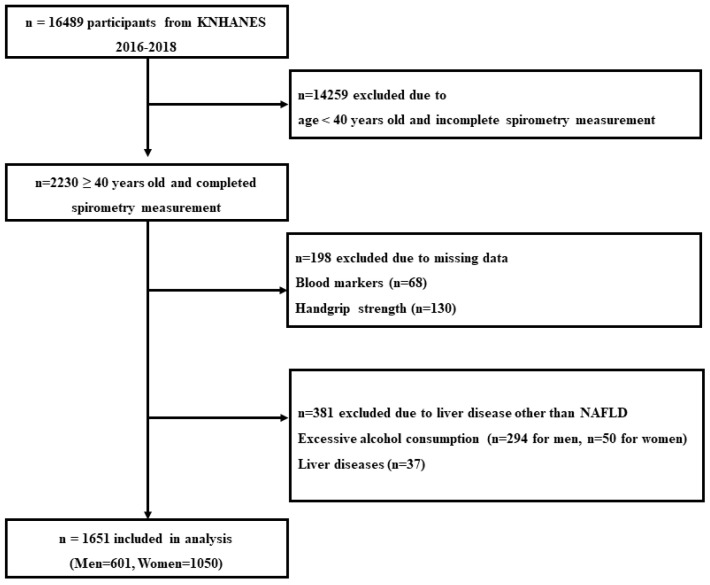
Flowchart of participants.

**Table 1 jcm-11-04151-t001:** Measured parameter values for NAFLD status by HSI.

Variables	Total (*n* = 1651)	Normal (*n* = 1230)	NAFLD (*n* = 421)	*p*-Value
Anthropometrics				
Age (years)	58.2 ± 11.4	58.0 ± 11.4	58.7 ± 11.5	0.244
Male, *n* (%)	601 (36.4)	451 (36.7)	150 (35.6)	0.725
BMI (kg/m^2^)	24.2 ± 3.3	22.9 ± 2.3	28.0 ± 2.9	<0.001
Relative HGS (kg/BMI)	1.25 ± 0.41	1.30 ± 0.40	1.10 ± 0.37	<0.001
Blood Markers				
AST (U/L)	22.8 ± 10.2	21.5 ± 9.1	26.5 ± 12.3	<0.001
ALT (U/L)	22.0 ± 15.3	18.2 ± 10.8	33.2 ± 20.3	<0.001
FBG (mg/dL)	103.8 ± 26.1	99.0 ± 17.7	117.7 ± 38.8	<0.001
TC (mg/dL)	195.6 ± 39.1	196.9 ± 37.3	191.6 ± 43.9	0.015
HDL-C (mg/dL)	50.3 ± 12.6	51.9 ± 12.8	45.7 ± 10.8	<0.001
TG (mg/dL)	147.9 ± 120.1	127.0 ± 104.9	172.7 ± 101.6	<0.001
SBP (mmHg)	121.6 ± 16.9	120.1 ± 16.8	126.0 ± 16.2	<0.001
DBP (mmHg)	75.8 ± 9.6	75.1 ± 9.6	78.1 ± 9.6	<0.001
CRP (mg/L)	1.27 ± 2.3	1.18 ± 2.36	1.52 ± 2.03	0.007
Health-Related Factors				
Smoking, *n* (%)	523 (32.1)	385 (31.8)	138 (32.9)	0.671
Alcohol consumption, *n* (%)	1012 (61.3)	783 (63.7)	229 (54.4)	0.001
Hypertension, *n* (%)	481 (29.1)	290 (23.6)	191 (45.4)	<0.001
Diabetes, *n* (%)	207 (12.5)	97 (7.9)	110 (26.1)	<0.001
Pulmonary Function				
FEV_1_ (%pred.)	90.7 ± 13.1	90.7 ± 13.3	90.7 ± 12.6	0.898
FEV_1_ (L)	2.5 ± 0.65	2.6 ± 0.65	2.5 ± 0.68	0.552
FVC (%pred.)	90.3 ± 12.6	91.5 ± 12.4	86.9 ± 12.3	<0.001
FVC (L)	3.3 ± 0.82	3.31 ± 0.81	3.20 ± 0.85	0.014
FEV_1_/FVC	0.78 ± 0.07	0.77 ± 0.07	0.79 ± 0.06	<0.001
Steatosis Index				
HSI	33.2 ± 4.8	31.0 ± 2.9	39.6 ± 3.2	<0.001

BMI: body mass index, WC: waist circumference, HGS: hand grip strength, AST: aspartate transaminase, ALT: alanine aminotransferase, FBG: fasting blood glucose, TC: total cholesterol, HDL-C: high-density lipoprotein cholesterol, TG: triglyceride, SBP: systolic blood pressure, DBP: diastolic blood pressure, CRP: C-reactive protein, FEV_1_: values of forced expiratory volume in 1 s, FVC: values of forced vital capacity, HSI: hepatic steatosis index.

**Table 2 jcm-11-04151-t002:** Measured parameter values for relative hand grip strength.

Variables	Total (*n* = 1651)	High HGS (*n* = 826)	Low HGS (*n* = 826)	*p*-Value
Anthropometrics				
Age (years)	58.2 ± 11.4	53.7 ± 9.4	62.7 ± 11.5	<0.001
Male, *n* (%)	601 (36.4)	301 (36.4)	300 (36.4)	0.507
BMI (kg/m^2^)	24.2 ± 3.3	22.9 ± 2.7	25.5 ± 3.23	<0.001
Relative HGS (kg/BMI)	1.25 ± 0.41	1.46 ± 0.37	1.03 ± 0.31	<0.001
Blood Markers				
AST (U/L)	22.8 ± 10.2	21.8 ± 10.1	23.7 ± 10.2	<0.001
ALT (U/L)	22.0 ± 15.3	20.6 ± 15.0	23.4 ± 15.5	<0.001
FBG (mg/dL)	103.8 ± 26.1	99.3 ± 20.2	108.3 ± 30.3	<0.001
TC (mg/dL)	195.6 ± 39.1	197.0 ± 38.0	194.1 ± 40.3	0.137
HDL-C (mg/dL)	50.3 ± 12.6	52.5 ± 12.6	48.1 ± 12.1	<0.001
TG (mg/dL)	147.9 ± 120.1	127.4 ± 117.4	149.9 ± 91.6	<0.001
SBP (mmHg)	121.6 ± 16.9	118.4 ± 15.8	124.8 ± 17.3	<0.001
DBP (mmHg)	75.8 ± 9.6	76.2 ± 9.3	75.5 ± 10.0	0.160
CRP (mg/L)	1.27 ± 2.3	1.05 ± 2.0	1.48 ± 2.5	<0.001
Health-Related Factors				
Smoking, *n* (%)	523 (32.1)	259 (31.5)	264 (32.7)	0.314
Alcohol consumption, *n* (%)	1012 (61.3)	583 (70.6)	42.4 (52.0)	<0.001
Hypertension, *n* (%)	481 (29.1)	138 (16.7)	343 (41.6)	<0.001
Diabetes, *n* (%)	207 (12.5)	63 (7.6)	144 (17.5)	<0.001
Pulmonary Function				
FEV_1_ (%pred.)	90.7 ± 13.1	91.4 ± 11.9	90.0 ± 14.3	0.031
FEV_1_ (L)	2.5 ± 0.7	2.8 ± 0.6	2.3 ± 0.6	<0.001
FVC (%pred.)	90.3 ± 12.6	93.3 ± 11.3	87.6 ± 13.1	<0.001
FVC (L)	3.3 ± 0.8	3.5 ± 0.8	3.0 ± 0.8	<0.001
FEV_1_/FVC	0.78 ± 0.07	0.78 ± 0.07	0.77 ± 0.08	0.001
Steatosis Index				
HSI	33.2 ± 4.8	31.6 ± 4.2	34.8 ± 4.8	<0.001

BMI: body mass index, WC: waist circumference, HGS: hand grip strength, AST: aspartate transaminase, ALT: alanine aminotransferase, FBG: fasting blood glucose, TC: total cholesterol, HDL-C: high-density lipoprotein cholesterol, TG: triglyceride, SBP: systolic blood pressure, DBP: diastolic blood pressure, CRP: C-reactive protein, FEV_1_: values of forced expiratory volume in 1 s, FVC: values of forced vital capacity, HSI: hepatic steatosis index.

**Table 3 jcm-11-04151-t003:** The risk of reduced pulmonary function according to NAFLD and relative HGS.

	OR (95% CI)	*p*-Value		OR (95% CI)	*p*-Value
FVC (%pred.)			FEV_1_ (%pred.)		
Normal	1		Normal	1	
NAFLD	1.938 (1.48–2.54)	<0.001	NAFLD	0.952 (0.71–1.28)	0.747
High HGS	1		High HGS	1	
Low HGS	1.951 (1.69–2.26)	<0.001	Low HGS	1.507 (1.16–1.96)	0.002

The risk of reduced pulmonary function (FVC < 80%, FEV_1_ < 80%) for normal vs. NAFLD and high HGS vs. low HGS. OR: odd ratio, CI: confidence interval, FEV_1_: values of forced expiratory volume in 1 s, FVC: values of forced vital capacity, NAFLD: non-alcoholic fatty liver disease, HGS: hand grip strength.

**Table 4 jcm-11-04151-t004:** The risk of reduced FVC according to NAFLD and relative hand grip strength-based subgroups.

	Crude Model		Model ^1^		Model ^2^		Model ^3^	
	OR (95% CI)	*p*-Value	OR (95% CI)	*p*-Value	OR (95% CI)	*p*-Value	OR (95% CI)	*p*-Value
Total (*n* = 1651)								
Normal + High HGS	1		1		1		1	
NAFLD + High HGS	1.297 (0.67–2.50)	0.436	1.443 (0.74–2.81)	0.281	1.455 (0.75–2.84)	0.271	1.398 (0.70–2.76)	0.341
Normal + Low HGS	3.062 (2.46–4.83)	<0.001	1.990 (1.38–2.87)	<0.001	2.017 (1.39–2.92)	<0.001	2.036 (1.40–2.97)	<0.001
NAFLD + Low HGS	4.489 (3.43–7.09)	<0.001	3.606 (2.46–5.29)	<0.001	3.581 (2.43–5.27)	<0.001	3.390 (2.23–5.15)	<0.001
Non-obese (*n* = 1042)								
Normal + High HGS	1		1		1		1	
NAFLD + High HGS	1.923 (0.42–8.73)	0.397	1.881 (0.41–8.69)	0.419	1.937 (0.42–8.97)	0.398	1.674 (0.35–8.02)	0.519
Normal + Low HGS	4.055 (2.62–6.27)	<0.001	2.339 (1.44–3.79)	0.001	2.454 (1.51–4.00)	<0.001	2.480 (1.51–4.07)	<0.001
NAFLD + Low HGS	6.396 (2.72–15.07)	<0.001	4.160 (1.70–10.16)	0.002	4.274 (1.74–10.48)	0.002	3.856 (1.49–9.99)	0.005
Obese (*n* = 609)								
Normal + High HGS	1		1		1		1	
NAFLD + High HGS	1.251 (0.68–2.31)	0.473	1.619 (0.85–3.10)	0.146	1.550 (0.80–2.99)	0.285	1.536 (0.78–3.01)	0.211
Normal + Low HGS	2.550 (1.36–4.77)	0.003	1.533 (0.78–3.01)	0.215	1.457 (0.73–2.90)	0.191	1.398 (0.68–2.86)	0.358
NAFLD + Low HGS	2.568 (1.46–4.51)	0.001	2.247 (1.23–4.11)	0.008	2.289 (1.25–4.19)	0.007	2.192 (1.15–4.16)	0.017

OR: odd ratio, CI: confidence interval, NAFLD: non-alcoholic fatty liver disease, HGS: hand grip strength; crude model: unadjusted; model ^1^: adjusted for age, sex; model ^2^: adjusted for age, sex, smoking; model ^3^: adjusted for age, sex, smoking, FBG, HDL-C, TG, SBP, DBP, CRP, alcohol consumption.

## Data Availability

The data from the KNHANES are available by visiting the Korea National Health and Nutrition Examination Survey website through the following URL: https://knhanes.kdca.go.kr/knhanes/sub03/sub03_02_05.do. These data can obtain free of charge for academic research.

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
