# Peer review of "Greater Muscular Strength Is Associated with a Lower Risk of Pulmonary Dysfunction in Individuals with Non-Alcoholic Fatty Liver Disease"

_jcm, 2022, doi:10.3390/jcm11144151_

Round 1
Reviewer 1 Report
In their comprehensive study of a large cohort Cho et al. investigated the relationship between handgrip strength (HGS) and NAFLD on pulmonary function.
To this end, the authors analyzed recorded data of forced vital capacity (FVC), forced expiratory volume in 1 sec (FEV1), handgrip strength (HGS) and the hepatic steatosis index (HSI) obtained from 1,651 subjects of two nationwide korean cross-sectional surveys, namely Korean National Health and Nutrition Examination Survey (KNHANES), from 2016 to 2018.
The authors observed, that NAFLD was inversely and HGS was positively associated with pulmonary function, assessed by forced vital capacity (FVC) measurement.
Based on their findings the authors conclude that NAFLD attenuates pulmonary function, while a higher HGS can reduce the risk of reduced pulmonary function.
While the article is of interest several aspects should be revised.
Introduction:
ll. 40
The authors cited the study from Paik and colleagues and stated that "an epidemiologic observation reported that pulmonary disease is one of the major ‘cause-specific death’ and ‘contributory causes of death’ in NAFLD [5]."
In order to better introduce the reader to the topic, it should be described in detail, which specific pulmonary diseases can be observed more frequently in NAFLD patients, which pulmonary diseases NAFLD patients die of and how this relationsship can be explained pathophysiologically.
What are the reason for the altered lung mechanis and altered gas exchange ?
Is there often a restrictive ventilatory disorder due to concomitant obesity ?
Are pulmonary processes favored by circulating elevated levels of proinflammatory cytokines in NAFLD?
The present explanations are too superficial and need to be described in more detail.
Methods:
Defintion of NAFLD:
The authors used the hepatic steatosis index (HSI) to define NAFLD.
Why was the HSI chosen and preferred over other systems such as the Fatty Liver Index (FLI) ? The justification that the HSI has been already established in the Korean population is not sufficient as a rationale alone from the reviewer's perspective.
Is there any further information on the severity of NAFLD ? Are data on fibrosis available ? Is there a non-invasive fibrosis measurement, e.g. by transient elastography or shear wave elastography ? Data on NAFLD Fibrosis Score (NFS) or FIB4 should be added.
HGS:
Information on frailty status including frailty index should be added.
Statistical analysis:
Which normality test was used ?
Results:
The authors reported, that the FVC (%pred.) value was significantly lower in the NAFLD than in the normal group (p<0.001)but there was no difference in FEV1 (%pred.) between the NAFLD and the normal (p>0.05).
A decrease in FVC is indicative of a restrictive ventilatory disorder.
Is this FVC reduction in the underlying NAFLD cohort because the NAFLD cohort had a significantly higher body weight and therefore a restrictive ventilatory disorder due to obesity?
A distinction should be made between lean and obese NAFLD patients and then assessed whether the association with reduced FVC persists or applies only to the obese subjects with increased body weight and abdominal circumference.
Information on the Tiffeneau index (FEV1/FVC ratio) should be added.
General points of criticism:
The authors repeatedly use terms such as "the NAFLD" or "the normal" within the manuscript.
This should be adapted to
"the NAFLD cohort" or "the NAFLD patients" or "in the patients with NAFLD".
respectively in
"the normal cohort" or "the normal patients" or "in the patients without NAFLD".
The authors write within the abstract or discussion that patients with NAFLD have impaired lung function or pulmonary dysfunction. This term ist o general. To be as precise and accurate as possible, it should be described exactly what was changed in the patients. In this case, the FVC.
Author Response
Thank you for the opportunity to respond to the reviewer comments regarding our manuscript, “Greater muscular strength is associated with a lower risk of pulmonary dysfunction in individuals with non-alcoholic fatty liver disease” [JCM-1770389].
We attached the response file.
Please see the attachment.

Reviewer 2 Report
I congratulate the authors on writing such a nice study regarding the importance of hand grip strength of the pulmonary function. It has been shown in many previous studies that hand grip strength is correlated with NAFLD and that NAFLD is associated with low pulmonary function
(doi: 10.1016/j.diabet.2019.04.008. Epub 2019 May 5). The highlight of the paper is combining hand grip strength + NAFLD + Pulmonary function. However, I do have some major concerns
1) the age gap between the Low HGS and high HGS is significant. This is important as age has been known to reduce muscle strength. So the low HGS group was already at a disadvantage as they would have a low FEV and FVC regardless of the NAFLD status as seen in table 4 results as well. How can you compare these groups when they are not age-matched. The groups for the HSI are age-matched and those results can be compared. This limitation should at least be mentioned in the limitations.
2) Model was used to predict the risk of reduced FVC in NAFLD, low HGS, and high HGS. Even though FEV correlated with low HGS, why wasn't the same model used to calculate the reduced FEV in the Low and High HGS ? It would be nice to see reduced FEV correlated with low Hand grip strength when adjusted with age.
3) The calculation is used to calculate the predicted values for FEV and FVC. I would like you to please add them here as I opened the reference and the paper was in Korean and could not understand the equation used. Please add it as you have done for calculating the HSI.
Author Response

(The authors gave the same response as above.)
